# Locally Recurrent Rectal Cancer According to a Standardized MRI Classification System: A Systematic Review of the Literature

**DOI:** 10.3390/jcm11123511

**Published:** 2022-06-18

**Authors:** Zena Rokan, Constantinos Simillis, Christos Kontovounisios, Brendan Moran, Paris Tekkis, Gina Brown

**Affiliations:** 1Department of Surgery and Cancer, Imperial College London, London SW7 2AZ, UK; constantinos.simillis@addenbrookes.nhs.uk (C.S.); paris.tekkis@rmh.nhs.uk (P.T.); gina.brown@imperial.ac.uk (G.B.); 2Pelican Cancer Foundation, Basingstoke RG24 9NN, UK; brendan.moran@hhft.nhs.uk; 3Cambridge Colorectal Unit, Addenbrookes Hospital, Cambridge CB2 0QQ, UK; 4Royal Marsden NHS Foundation Trust, London SW3 6JJ, UK; 5Chelsea & Westminster Hospital, London SW10 9NH, UK; 6Basingstoke & North Hampshire Hospital, Basingstoke RG24 9NA, UK

**Keywords:** locally recurrent rectal cancer (LRRC), rectal cancer, BTME classification

## Abstract

(1) Background: The classification of locally recurrent rectal cancer (LRRC) is not currently standardized. The aim of this review was to evaluate pelvic LRRC according to the Beyond TME (BTME) classification system and to consider commonly associated primary tumour characteristics. (2) Methods: A systematic review of the literature prior to April 2020 was performed through electronic searches of the Science Citation Index Expanded, EMBASE, MEDLINE, and CENTRAL databases. The primary outcome was to assess the location and frequency of previously classified pelvic LRRC and translate this information into the BTME system. Secondary outcomes were assessing primary tumour characteristics. (3) Results: A total of 58 eligible studies classified 4558 sites of LRRC, most commonly found in the central compartment (18%), following anterior resection (44%), in patients with an ‘advanced’ primary tumour (63%) and following neoadjuvant radiotherapy (29%). Most patients also classified had a low rectal primary tumour. The lymph node status of the primary tumour leading to LRRC was comparable, with 52% node positive versus 48% node negative tumours. (4) Conclusions: This review evaluates the largest number of LRRCs to date using a single classification system. It has also highlighted the need for standardized reporting in order to optimise perioperative treatment planning.

## 1. Introduction

Rectal cancer is common and accounts for almost 30% of all cancers of the colon and rectum. Surgery can be technically difficult due to the narrow confines of the pelvis and the proximity of vital adjacent organs. For these reasons, locally recurrent rectal cancer (LRRC) may commonly be a result of the progression of a residual disease and thus a potentially preventable condition. LRRC manifests in approximately 5–18% of patients after surgery [1,2,3], and if untreated can lead to significant morbidity. The optimal outcome in rectal cancer surgery is complete oncological clearance of the tumour (R0 resection), delivered safely to patients suitable for surgery, as this has been reported as the best predictor of disease free and overall survival [1]. Achieving this is dependent on optimal surgery largely guided by accurate pre-operative imaging. There is now almost universal agreement that, where feasible, pelvic MRI is the gold standard imaging modality for the assessment of both primary rectal cancer and more crucially for staging local recurrence (LR) [4].

Currently there is no single classification system used to describe and categorise LRRC, although multiple anatomical and operative systems have been proposed [5,6,7,8,9,10,11,12,13]. Members of the Beyond TME Consensus Group [4] have devised an MRI classification, which describes the anatomical tumour location within one of seven intra-pelvic compartments (The BTME Classification) (Table 1). Compartments are formed by fascial boundaries along the potential planes of dissection (Figure 1, Figure 2 and Figure 3).

Georgiou et al. previously validated this classification system by implementing MRI staging pre-operatively to plan surgical resection and assess oncological and survival outcomes. They demonstrated that patients with a tumour within the ‘anterior above peritoneal reflection’ compartment on MRI had a worse overall survival compared with patients where this compartment was not involved (*p* = 0.012) [5]. They also reported that patients with a tumour within the lateral and posterior compartments, or within three or more compartments had a reduced disease-free survival [14].

**Figure 1 jcm-11-03511-f001:**
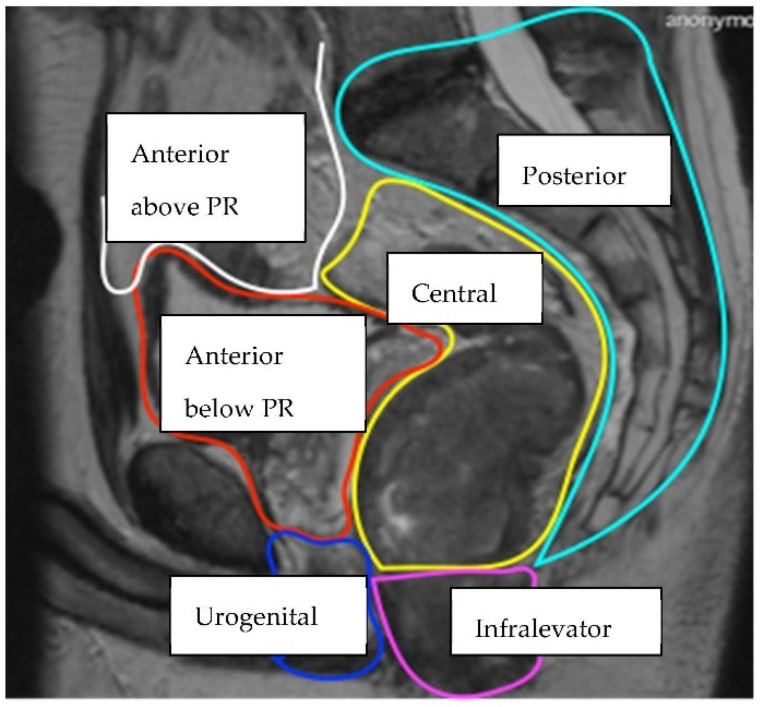
Sagittal MRI view of the intra-pelvic compartments (Key: PR = peritoneal reflection) [15].

**Figure 2 jcm-11-03511-f002:**
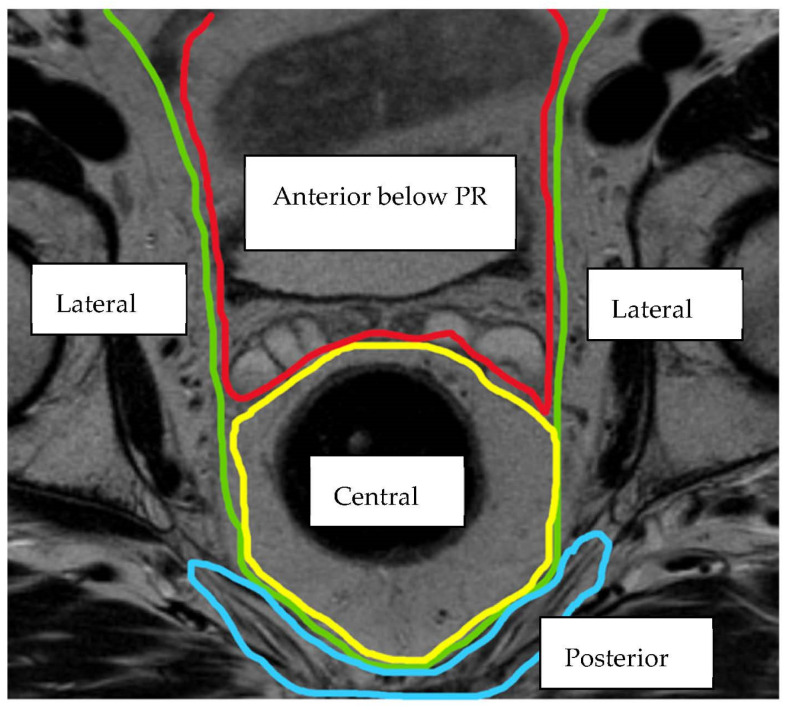
Axial MRI view of the intra-pelvic compartments (Key: PR = peritoneal reflection) [15].

**Figure 3 jcm-11-03511-f003:**
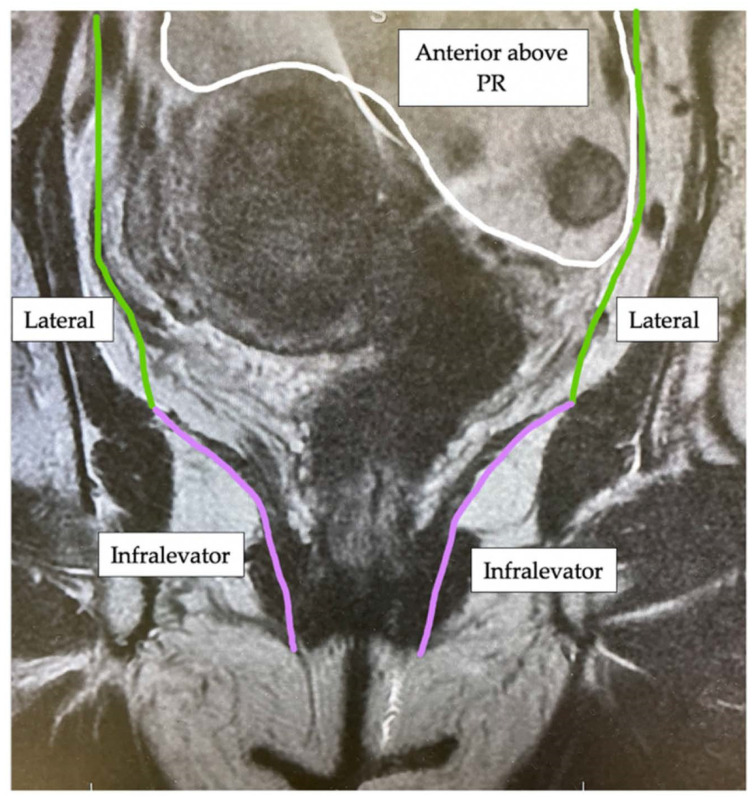
Coronal MRI view of the intra-pelvic compartments (Key: PR = peritoneal reflection).

The aim of this review is to assess from the literature, the location and frequency of LRRC following previous surgery for primary rectal adenocarcinoma, using an established classification system, in this case the BTME system. This volume of information has not previously been reported in such a standardized manner, and as a result we aim to understand the common sites of recurrence in order to understand the most frequent operations that are required for these recurrences and where the emphasis should be on resource utilisation in the future.

## 2. Materials and Methods

### 2.1. Search Strategy

This systematic review was based on a written protocol and was reported in line with Preferred Reporting Items for Systematic Reviews and Meta-Analyses (PRISMA) [16] and Assessing the Methodological Quality of Systematic Reviews (AMSTAR) guidelines [17]. A comprehensive literature search was performed using a combination of free-text terms and controlled vocabulary of the following databases: PubMed MEDLINE, Embase, Science Citation Index Expanded, and Cochrane Central Register of Controlled Trials (CENTRAL) in The Cochrane Library. The detailed search strategy is provided in Appendix A.

All abstracts, studies, and citations identified were reviewed, and references in the identified studies were also searched. No restrictions were made based on language, publication year, or publication status. The literature search was complete up to 28 April 2020.

### 2.2. Selection Criteria

Prospective and retrospective studies were considered for this systematic review if studies met the following criteria:Reported on patients with LRRC or rectosigmoid cancer who underwent previous ‘curative’ surgery (R0 resection).Reported on patients where the anatomical location of LR or an established classification system for describing LRRC was documented.

### 2.3. Outcomes of Interest

#### 2.3.1. Primary Outcome

The primary outcome was to assess the frequency and location of LRRC within the pelvis when the BTME classification system was applied to studies, which have previously classified LRRC according to an established system or provided an anatomical location of the LR within the pelvis.

#### 2.3.2. Secondary Outcomes

To evaluate LRRC by applying the BTME system where applicable, in relation to:Height of the primary tumour;The primary surgical procedure performed;Tumour node metastasis (TNM) staging, extramural vascular invasion (EMVI), and nodal status of the primary tumour;Perioperative treatment received for the primary tumour.

Two review authors (ZR and CS) independently determined the eligibility of all retrieved studies and extracted the required data from the included studies.

## 3. Results

### 3.1. Study Stratification

#### 3.1.1. Eligible Studies

A total of 3908 references were identified through systematic electronic searches of Science Citation Index Expanded (*n* = 1140), EMBASE (*n* = 1091), MEDLINE (*n* = 1563), and CENTRAL (*n* = 114). A further 29 studies were identified from manuscripts referenced in the above studies. There were 2017 duplicates between databases, which were excluded. A further 1816 clearly irrelevant references were excluded through screening titles and reading abstracts. The remaining 230 studies were investigated in full text detail and a further 172 studies were excluded. The reasons for exclusion included conference abstract papers, no English version or ability to translate the paper, cancers only classified as ‘locally recurrent’ within the paper, or those classifying anal or prostate cancer. Appendix A. shows the study flow diagram. Fifty-eight cohort studies fulfilled the inclusion criteria of this systematic review [6,7,9,11,12,13,18,19,20,21,22,23,24,25,26,27,28,29,30,31,32,33,34,35,36,37,38,39,40,41,42,43,44,45,46,47,48,49,50,51,52,53,54,55,56,57,58,59,60,61,62,63,64,65,66,67,68,69]. The characteristics of these studies including patient demographics, primary tumour staging, and treatment received are summarised in Appendix A.

#### 3.1.2. Included Studies

Overall, 58 studies, were identified, of which 19 were prospective and 39 retrospective cohort studies. In the 58 studies, 3975 patients with LRRC were identified. These recurrences often occupied more than one anatomical location/compartment and, therefore, 4558 sites of LRRC were included for classification. All studies categorised LRRC purely by anatomical location or by an established system outlining a regional/compartmental anatomical location, a degree of fixation within the pelvis, or symptoms associated with LRRC.

In 21/58 studies, LRRC’s were reported according to a previously established classification system [6,7,9,11,12,13,24,31,34,38,39,40,51,54,56,57,59,65,66,67,68], however, in the majority (37/58) of studies, LRRC was described according to an anatomical/regional location.

### 3.2. Location

The location of each LRRC was classified using the BTME system (Figure 4) with the exception of 369 patients who were classified only according to fixity as per the Mayo Clinic system [6] and, therefore, compartmental location could not be assessed (Figure 5).

The remaining 4189 sites of LRRC, were grouped as follows: central (18%), lateral (15%), posterior (13%), anterior below peritoneal reflection (10%), infralevator/anterior urogenital triangle (4%), and anterior above peritoneal reflection (1%). The infralevator and anterior urogenital compartments were grouped together for the purposes of this review as from the published literature, LR within the perineum was difficult to define.

The precise location of 1621/4189 (39%) sites could not be categorised into these individual compartments as it was impossible to extract the appropriate information from the published results. For example, the anatomical description may have been too vague, including ‘pelvic’, ‘non-central’, or ‘retroperitoneal’ LRRC.

### 3.3. Height of Primary Tumour

Details on the height of the primary tumour according to the distance from the anal verge, was available for 1873/3975 (47%) patients with LRRC. Of these, 703/1873 patients had a primary tumour classified only as ‘below the peritoneal reflection’. The remaining 1170 tumours could be grouped into either upper/lower rectum (91 patients) or into thirds as upper/middle/lower rectum (1079 patients). With the rectum divided into thirds, 23% of patients had an upper rectal primary tumour, 40% of patients a middle rectal tumour, and 37% of patients a lower rectal tumour. When information was provided for the rectum divided only into upper and lower, 38% of patients had an upper rectal primary tumour and 62% of patients a lower rectal primary tumour.

### 3.4. Primary Surgery

Information on the primary surgical procedure was available for 3254/3975 (82%) patients with LRRC (Figure 6). The most frequent operation was a restorative anterior resection (AR) in 44%, abdominoperineal excision of the rectum (APER) in 33%, local excision (LE) in 7%, and Hartmann’s procedure in 1% of patients. In 15% of patients, the procedure was categorised as ‘other’ as the information provided in the original study was indistinct, i.e., ‘TME’ or ‘sphincter-sparing resection’, which could potentially incorporate both restorative and non-restorative operations. This category also included operations, such as rectal stump excision and subtotal colectomy.

In 313 of these patients, information on the primary surgical procedure in addition to the anatomical location of the subsequent LR was available and Figure 7 outlines this classified according to the BTME system.

Following both anterior resection and LE, LR was most frequent in the central pelvic compartment (47/130 and 84/103, respectively), compared to the infralevator/anterior urogenital triangle following APER (32/80). These two compartments were again amalgamated.

### 3.5. TNM Stage

In 2857/3975 (72%) patients (40/58 studies), staging information on the primary tumour was provided. As staging systems have evolved over time, these included: Tumour Node Metastasis (TNM), Japanese Society for Cancers of the Colon and Rectum (JSCCR), Dukes’, Astler-Coller (also modified), and American Joint Committee on Cancer/Union for International Cancer Control (AJCC/UICC) classifications (Figure 8).

The majority (1814/2857, 63%) of these patients had a more ‘advanced’ primary when staged within the different systems: TNM T3–T4 (1163/1552, 75%), JSCCR ‘a1’ (5/9, 55%), Dukes’ C & D (286/513, 56%), Astler-Coller ‘C’ (162/381, 43%), and AJCC/UICC Stage III/IV (198/402, 49%). Four patients had their primary cancer staged as ‘T0’ following chemoradiotherapy or ‘Tis’ following invasive cancer excised by polypectomy.

### 3.6. EMVI Status

EMVI status was only documented in 8/58 (14%) studies, which was 1454/975 (38%) of patients. Of the 1454 patients in which EMVI status was available, 482/1454 (33%) were EMVI positive and 972/1454 (67%) EMVI negative.

### 3.7. Nodal Status

Information was available on the nodal status of the primary tumour in 2549/3975 (64%) patients. The majority of patients, 1320/2549 (52%), were node positive, and 1229/2549 (48%) patients were node negative. Of those node positive patients with more detailed information, 173/1320 patients were staged as N1 and 118/1320 patients as N2.

In 63/2549 patients the location of LR in addition to nodal status of the primary tumour was available (Figure 9). The majority of these were lateral LR’s and of note, none of these patients underwent lateral lymph node dissection at the primary surgery.

### 3.8. Perioperative Treatment

Information on the perioperative treatment received for the primary tumour was only available for 982/3975 (25%) patients who developed LRRC (Figure 10). Of these, 296/982 patients received neoadjuvant radiotherapy, 242/982 adjuvant chemotherapy, 152/982 neoadjuvant chemoradiotherapy, 97/982 adjuvant radiotherapy, 92/982 neoadjuvant/adjuvant chemoradiotherapy, 30/982 radiotherapy, 24/982 neoadjuvant chemotherapy, 17/982 adjuvant chemoradiotherapy, 17/982 neoadjuvant ‘treatment’ (chemotherapy/radiotherapy/chemoradiotherapy), 13/982 adjuvant ‘treatment’, and 2/982 chemotherapy.

## 4. Discussion

Although the rates of LRRC have significantly reduced following the advancement of treatment in patients with primary rectal cancer, the burden of this pathology remains. This review has highlighted some of the risk factors associated with LRRC and the patterns of recurrence following surgery for primary rectal cancer.

Success with respect to surgery for LRRC is based upon achieving complete oncological clearance (R0 resection), which in turn is associated with improved survival outcomes [1]. This review paper is an observational snapshot of outcomes following rectal cancer surgery, resulting in LR, which has been reported according to a standardized system. By understanding the frequency and locations of LRRC, primary tumour characteristics commonly resulting in LR and operations most frequently performed in this complex cohort of patients, we are better informed with regards to treatment planning in the future and as a result, more likely to achieve R0 resection. Familiarising multidisciplinary clinicians with this information also enables optimisation of services and appropriate resource delegation.

Use of imaging for the assessment of LRRC has evolved over time, with MRI now being regarded as the gold standard modality [4]. Postoperative imaging surveillance following primary rectal cancer surgery is predominantly CT-based, however, on suspicion of LR, MRI and also PET-CT provide increased soft tissue contrast in comparison to CT alone [70]. MRI was stated as the main diagnostic tool in only 5/58 studies within this review, which is unsurprising considering that many of the included studies were written prior to its development and widespread implementation. It must also be mentioned that, while many of the studies did not explicitly state a single imaging modality used to diagnose recurrence, this did include the use of MRI. The BTME classification is MRI-based, although all of the systems described could easily be adapted to the progression in imaging.

As there is currently no single system in use for the classification of LRRC, within this review we have categorised those from the published literature according to the BTME system, which has resulted in the largest number of studies and reported LRs classified according to a single system. However, as mentioned, other anatomical and operative classifications are in use, some similar to the BTME system and all beneficial in providing a consistent manner to describe and radiologically report on LRRC. Use of a standardized system at least at a local level, if not wider, allows comparison of oncological outcomes based on a radiological assessment of the primary tumour. This in turn should not only improve information provided to patients with regard to their prognosis at primary diagnosis, but also provide a format on which audit and research can be based.

Oncological clearance and the prevention of disease recurrence following surgery for primary rectal cancer was most significantly influenced by the introduction of TME surgery by Professor Richard J. Heald. By removing the lymphovascular tissue responsible for cancer spread within the mesorectum during 132 consecutive ‘curative’ anterior resections, Professor Heald demonstrated a reduction in the rate of LR to three patients within 9 years (2.3%) [71]. LRRC, however, is likely to be multifactorial and influenced by a number of different factors associated with the primary tumour, including the surgical procedure performed, height from the anal verge, EMVI status, nodal status, and TNM stage. Circumferential Resection Margin (CRM) status is also incredibly important in assessing the risk of LR [72]. Within this review, the CRM status of the primary tumour was mentioned in 10/58 (17%) studies [20,21,22,26,32,33,38,40,51,65], however, data were often incomplete or the CRM status attributed to the LRs could not be defined, therefore, these small numbers were not formally assessed. All patients within this review were, however, deemed to have had complete oncological clearance of the primary tumour.

### 4.1. Location

When reviewing the location of LRRC within the pelvis, the largest proportion occurred within the central compartment (757 LRs) followed by the lateral compartment (610 LRs). Large numbers of LRs within this study were also located in the posterior and anterior below peritoneal reflection compartments, with much lower numbers or no LRs within the anterior above, anterior urogenital, and infralevator compartments. Some of the classification systems used within these studies group LRs together for example, and those infiltrating the prostate or vagina or anterior to the rectum above the peritoneal reflection, as located within the central compartment. A limitation of this review is that by dividing this central compartment into BTME sub-compartments according to the fascial planes, the numbers are most likely reduced in these compartments as the majority of studies have not stratified LR according to this level of detail.

*Why is recurrence within the central compartment most prevalent?* It has been suggested that anastomotic recurrences result from intraoperative tumour spillage or remnant tumour cells during excision of the primary tumour and that mesorectal LRs occur subsequent to an incomplete/partial TME, prior to TME becoming the standard procedure for excision of rectal tumours [10]. Publication dates within this review ranged from between 1960 to 2020, with TME surgery first being described back in 1982 [73]; however, it took many years thereafter for TME surgery to become standard practice. However only 84/757 (11%) of the included LRs occurring within the central compartment were prior to 1990 [11,19,43,44,50,58,69]. Higher rectal/rectosigmoid cancers may have resulted in central LRs due to partial TME, essentially tumours not requiring a full TME and, therefore, leaving some of the lymphovascular tissue behind. It has also been suggested that anastomotic leakage following anterior resection increases the risk of LR in a similar manner. Extraluminal leakage of free malignant cells from the anastomosis results in LR within the pelvis [74], typically within the central compartment.

Lateral and posterior LRs followed central recurrences in prevalence. Rectal tumours over 5 cm from the anal verge treated with neoadjuvant radiotherapy or chemoradiotherapy are associated with a higher frequency of presacral and lateral LRs [75], which was also indicated within this review.

Beppu et al., on reporting the outcomes of T3 low rectal cancers, proposed that LR in the lateral compartment may be as a result of the direct spread through the mesorectum to lateral pelvic sidewall lymph nodes. Because these nodes are not apparent nor resected during TME dissection, they present as lateral LR [21]. A similar suggestion was also postulated by Fan et al. who proposed that retrieving an inadequate lymph node sample during TME may indicate an incomplete resection and adverse oncological outcomes [76].

Kusters et al. demonstrated that the 5-year LR rate was greater in those undergoing TME and unilateral lateral lymph node dissection compared to in those undergoing TME and bilateral lateral lymph node dissection for mesorectal lymph node positive rectal cancer (33% vs. 14%, *p* = 0.04) but also in all patients regardless of lymph node status (15% vs. 8%, *p* = 0.06). This has been suggested to be as a result of the tumour remaining in lateral lymph nodes during unilateral TME or in a standard TME operation. Tumour cells may also remain in the lateral lymph flow system during unilateral lymph node dissection and leak out causing LR not only laterally, but in the presacral, perineal, and anastomotic sites [77].

### 4.2. Height of Primary Tumour, Primary Surgery and T-Stage

The height of the primary tumour, particularly differentiating between high and low rectal tumours, in addition to the primary surgery performed, is well recognised as having a significant effect on LR rates.

Within this review, in patients having previously undergone a local excision or anterior resection for their primary cancer, LR was most prevalent in the central compartment. In those having undergone an APER, LR was most prevalent in the infralevator/anterior urogenital triangle, i.e., the perineum followed by the posterior and anterior below compartments. LR within the lateral compartment was appreciably more prevalent in those following anterior resection.

Hruby et al. found that in patients with LRRC in the anterior central compartment, a significant proportion of these had primary T4 tumours on histopathology (*p* < 0.01) and that perineal LRs occurred most significantly following APER (*p* < 0.01) compared to in no patients who had undergone a previous AR [13]. TME is now the universally accepted technique for excision of rectal cancer, however, even Professor Heald, having personally operated on 502 patients, determined that LR at both 5 and 10 years was more prevalent in those undergoing APER in comparison to anterior resection (17% & 36% vs. 5% and 5%, *p* < 0.001). He hypothesised that this may be as a result of an absent mesorectal/dissection plane below 4 cm from the anal verge and that the imprecision of TME dissection during an APER may result in damage to the specimen [78].

When Kusters et al. compared LR rates at 5 years following surgery for low rectal cancer between patients in the Netherlands and Japan, there was a significant difference in 5-year LR rates (12.1% vs. 6.9%), which was attributed in part to a wider perineal excision performed routinely in Japan [38], but these patients may also have had a lateral pelvic lymph node dissection, if indicated.

Results from the Dutch TME trial suggested that LR following APER occurs most frequently in the presacral area, which is suggested could be attributed to tumour spillage from positive resection margins implanting in the midline in the presacral area, due to natural gravitational forces [10,38]. It was also found that within this population of patients undergoing a conventional APER, the resection margin was located within the sphincter muscle or submucosa in over one-third of cases [79]. Therefore, for complete oncological clearance, the dissection plane may have to be adapted from that of a conventional APER to an interpshincteric APER or even an extralevator APER (ELAPE).

Within this review when assessing height of the primary tumour, the rectum was divided into halves or thirds. In either case, a low rectal tumour was defined as less than 5 cm or 6 cm from the anal verge, for which there were 457/1170 LRs with information on height of the primary tumour. Low rectal cancer has been associated with increased rates of recurrence compared with those higher up in the rectum. This is thought previously to have been contributed to by the anatomy of the lower rectum with ‘waisting’ of surgical TME specimens and also as a result of surgical technique. With the mesorectum tapering into a point at its insertion on the superior surface of puborectalis sling, excision distal to this can leave a tumour potentially exposed. A lack of surrounding mesorectal fat results in the sphincter muscle forming the circumferential resection margin (CRM) [79], with CRM positivity (tumour less than 1 mm from the CRM) being a strong predictor of LR [79,80,81]. This was emphasised in the MERCURY study, demonstrating that a positive CRM on MRI was an independently significant predictor of LR regardless of other preoperative factors [81].

Nagtegaal et al. on reporting on 1219 patients with clinically resectable rectal cancer within the Dutch TME trial indicated that the anatomical and surgical challenges faced by difficult access to low rectal tumours, resulted in a high CRM positivity rate. Regardless of whether either APER or anterior resection was performed for a low rectal tumour, local recurrence rates were higher in those with a positive CRM (APER *p* = 0.002 vs. AR *p* = 0.07) [80].

Räsänen et al. described 40 LRs in 481 patients following curative rectal cancer surgery and demonstrated an increased risk of LR in those with tumours below 6 cm from the anal verge compared with those above 6 cm (*p* = 0.005). They have suggested that this may be attributable to those undergoing APER having more extensive disease and conversely that those undergoing a Hartmann’s procedure may be subsequent to tumour perforation or in patients with a suspicion of an incomplete oncological resection [82].

### 4.3. Perioperative Treatment, EMVI and Nodal Status

This cohort represents patients deemed to have undergone curative resection defined as R0 on original pathology. It is assumed, but not proven, that these resections were CRM negative. Therefore, there remains a challenge to understand the causes and patterns of LR when patients have undergone curative surgery. This systematic review shows that neither LR nor the perioperative treatments associated with LR or EMVI status are consistently documented. This is of great concern given that nearly all pre-operative strategies are aimed at preventing pelvic LR and that EMVI status, is regarded as a predictor of local failure. Nodal status is also a strongly considered factor in determining perioperative treatment planning at most centres; however, results from this review have demonstrated that near equal proportions of patients have LR following the resection of their primary tumour, regardless of nodal status. This calls into question whether LR can be attributed to nodal status alone and may significantly impact treatment planning moving forward.

Where data are complete, it is clear that recurrences cannot be classified as simply central or lateral since the anatomy of recurrence within the pelvis is far more complex. The classification system of seven compartments was applied to the descriptions given in the published literature. Since the classification has not been in widespread use, we anticipated that we would have to amalgamate compartments where appropriate, for example, the pelvic floor should be divided into an anterior urogenital versus posterior hindgut compartment, separated by the perineal body. However, when the term ‘perineal’ recurrence is used, this often does not specify whether recurrence extends anteriorly, posteriorly or both. Therefore, the pelvic floor below the levator origin was amalgamated for the purposes of this review.

In future, a minimum standard for recording recurrences should be followed. This should include detailed documentation of the original primary tumour stage that includes T-stage, N-stage, EMVI status, and tumour height, as well as greater detail about the precise treatment given to the original primary tumour. Pre-operative MRI CRM status, as mentioned, has been demonstrated to be an independent predictor of LR, and, therefore, including this information in conjunction with these factors is imperative and should not be assumed.

## 5. Conclusions

Use of a single classification has enabled us to compare factors relating to the primary tumour between centres. In order to further assess this within a large volume of patients, radiological reporting in a consistent manner, by use of a standardized classification system, is essential. This will allow further audit and research to be performed within this field and the BTME system is just one of the available reporting systems that can be used. MRI is the gold standard imaging modality, which perhaps should be considered during routine surveillance at the very least in patients at high risk of LRRC, enabling prompt and accurate diagnosis.

## Figures and Tables

**Figure 4 jcm-11-03511-f004:**
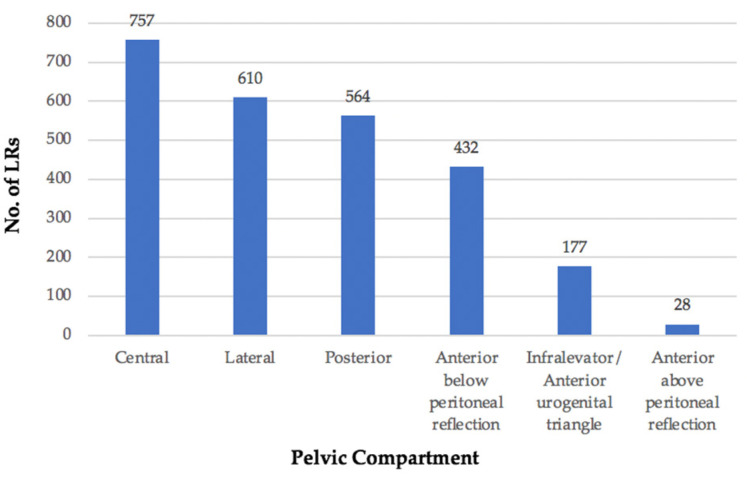
No. of LRs occupying each pelvic compartment.

**Figure 5 jcm-11-03511-f005:**
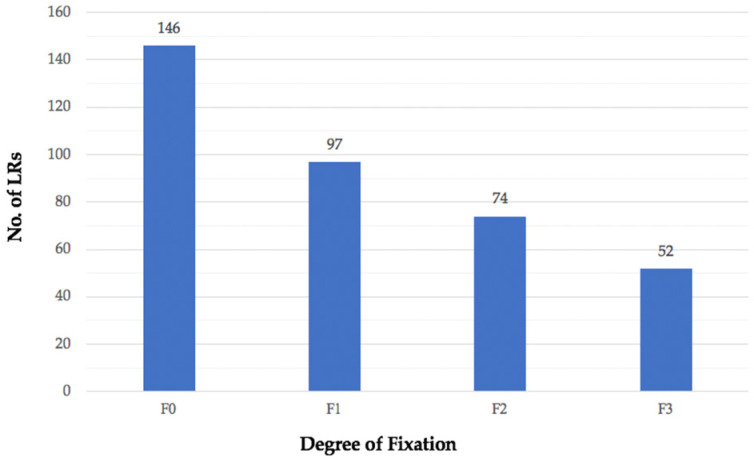
No. of LRs classified only according to degree of fixation to surrounding pelvic structures.

**Figure 6 jcm-11-03511-f006:**
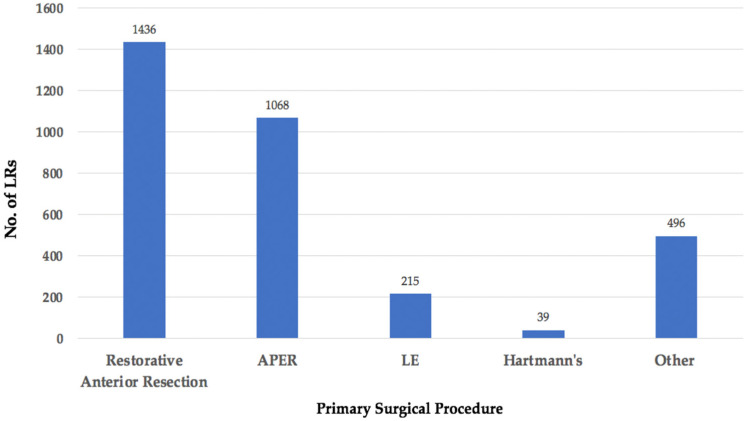
Primary surgical procedures resulting in LR.

**Figure 7 jcm-11-03511-f007:**
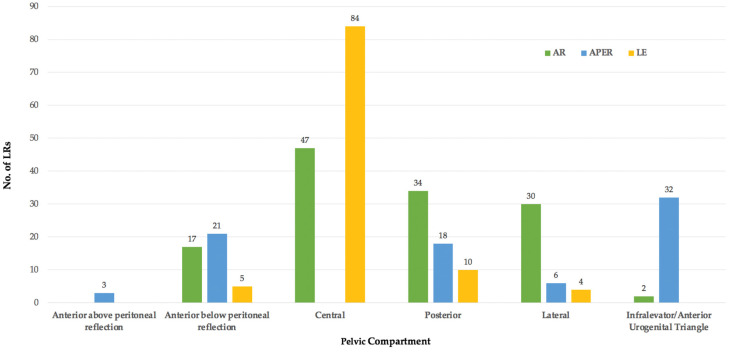
Compartmental location of LR according to operation performed for primary cancer.

**Figure 8 jcm-11-03511-f008:**
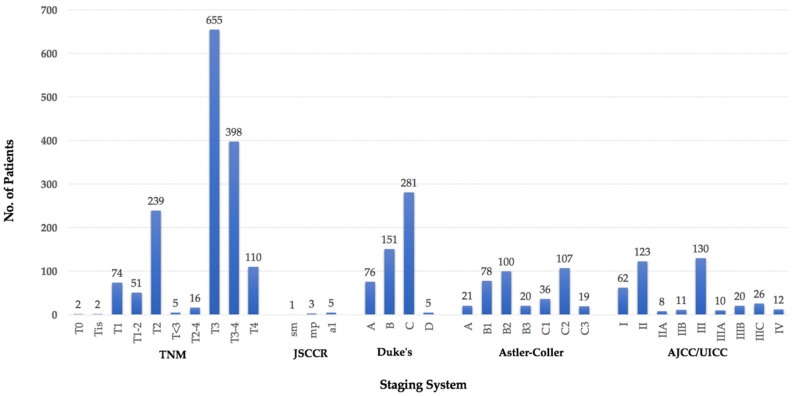
LRs according to staging of primary tumour.

**Figure 9 jcm-11-03511-f009:**
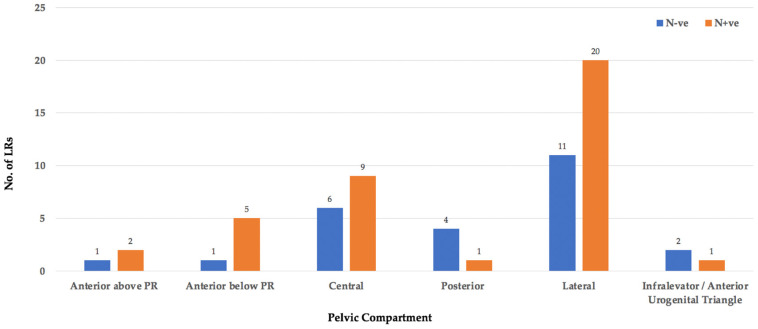
Nodal status leading to LR per pelvic compartment.

**Figure 10 jcm-11-03511-f010:**
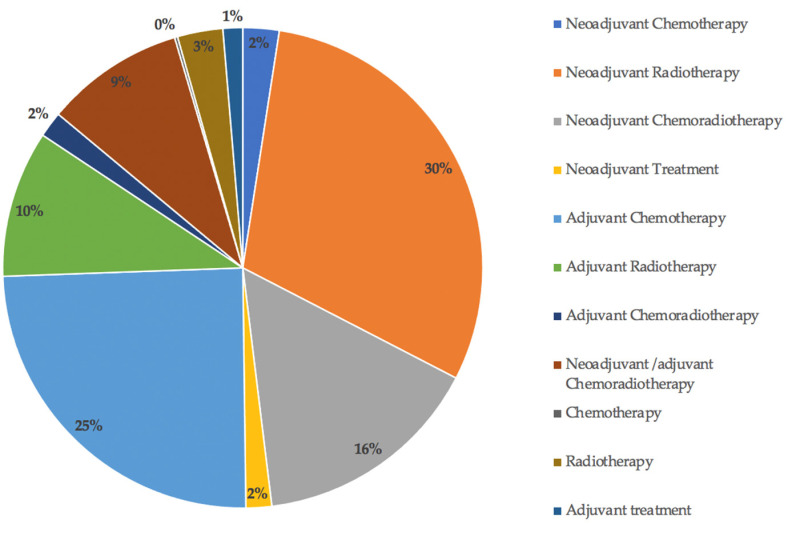
Perioperative treatment received for the primary cancer prior to the development of LR.

**Table 1 jcm-11-03511-t001:** BTME classification of intra-pelvic anatomical compartments.

Compartment	Structures within Compartment
**Anterior above Peritoneal Reflection**	Ureters, iliac vessels above peritoneal reflection, sigmoid colon, small bowel, lateral pelvic sidewall fascia (peritoneal surface)
**Anterior below Peritoneal Reflection**	Genitourinary system (seminal vesicles, prostate, uterus, vagina, ovaries, bladder/vesico-ureteric junction, proximal urethra), pubic symphysis
**Central**	Rectum/neo-rectum (intra/extra-luminal), perirectal fat or mesorectal recurrence
**Posterior**	Coccyx, pre-sacral fascia, retro-sacral space, sacrum, sciatic nerve, sciatic notch, S1 and S2 nerve roots
**Lateral**	Internal and external iliac vessels, lateral pelvic lymph nodes, piriformis muscle, internal obturator muscle
**Infralevator**	Levator ani muscles, external sphincter complex, ischio-anal fossa
**Anterior Urogenital triangle**	Perineal body/perineal scar (if previous abdomino-perineal resection of rectum), vaginal introitus, distal urethra, crus penis

## Data Availability

The data presented in this study are available in Appendix A.

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
