# Peer review of "Locally Recurrent Rectal Cancer According to a Standardized MRI Classification System: A Systematic Review of the Literature"

_jcm, 2022, doi:10.3390/jcm11123511_

Round 1
Reviewer 1 Report
The topic is interesting and the article is well-written.
I think that a sentence about the "grey literature" ( why the authors decide to not include) is necessary. Some sentences about strenghts and limitations of the systematic review may be useful to understand the main message of the review.
Author Response
Dear reviewer,
Many thanks for your constructive comments and for taking the time to review the article, it is very much appreciated. As per your points:
- We have inserted a sentence as you suggest and about the literature that was excluded and outlined the reasons for this under 'eligible studies' in section 3.1.1
- With regards to strengths and limitations - we feel the a strength of this study is the volume of data collated according to a single classification system. This provides a snapshot of some patterns of local recurrence and has highlighted the need for standardized reporting.
- With regards to limitations, we have stated that we amalgamated the infralevator and anterior urogenital compartments and that CRM status has not been assessed due to small numbers. The main message concerns standardized reporting of local recurrence and the need to consider important information regarding the primary tumour in future research to help predict patterns of recurrence.
We hope that addresses the comments and suggestions.
Reviewer 2 Report
Excellent article. I have no revisions to suggest.
Author Response
Thank you very much, we are very grateful to you for reviewing this article.
Reviewer 3 Report
First of all, I would like to congratulate the authors for their work. This paper titled “Locally Recurrent Rectal Cancer According to a Standardised MRI Classification System: A Systematic Review of the Literature” gives a perfect view of the current published data about Locally Recurrent Rectal Cancer (LRRC). In this paper we are able to understand some patterns of recurrence following primary rectal cancer resections. Initially, the authors summarize the frequency of the location of appearance of the LRRC. For the secondary outcomes, they do a thorough analysis of the published data, doing subanalysis in relation to the height of the primary tumor, the type of primary surgical procedure performed, the TNM stage, the EMVI and the nodal status of the primary tumor and the perioperative oncological treatment realized on the primary tumor.
As the current literature is very heterogenic, it’s clear that it’s mandatory to use some kind of standardised anatomic classification system to improve both prognosis for the patient and future investigation.
The manuscript is well written and structured. No grammatical adjustments are needed.
I have only some minor comments:
- Consider adding some more figures to help to understand the BTME classification system. Maybe a coronal image of the “anterior above PR” compartment would be helpful.
- In page 4/15, in the first paragraph you talk about EMVI. You should explain the acronym.
- When analyzing results between nodal status and LRRC, most of them are lateral recurrences. When describing the surgical technique I found no reference about lateral lymph node dissection. Is this data missing? If you have this data it should be included and if it’s missing it’s not a major problem (it’s a review of the literature) but you should mention it.
- On page 10, on the last line there are two “this”.
- In your review only in 17% studies the CRM status is mentioned. Therefore, the results were not assessed. You mention this aspect in the discussion and that it is important for future studies. Do you mean the preoperative CRM on MRI like in the MERCURY study? Or the pathological founds on the Dutch TME trial you mention? Or both of them? Please specify
Congratulations to the authors.
Author Response
Dear reviewer,
Many thanks for your kind review and very constructive comments. As per your points below:
- We have included a coronal figure as you suggest.
- We specify that EMVI is extramural vascular invasion
- Under section 3.7 we have inserted a sentence regarding lateral lymph node dissection. In the patients where primary nodal status and resultant lateral local recurrence is documented, none of these had undergone a lateral lymph node dissection at the primary operation. We have outlined the main operations performed
- Many thanks, we have corrected this
- Thank you. This refers to pre-operative CRM status on MRI. We have included a sentence to this effect.
We hope that these alterations are sufficient and that the manuscript is now suitable for publication.
Many Thanks